# DEEP NETWORK PARTITION DENSITY EXHIBITS DOUBLE DESCENT

## ABSTRACT

The study of Deep Network (DN) training dynamics has largely focused on the evolution of the loss function, evaluated on or around train and test set data points. In fact, many DN phenomenon were first introduced in literature with that respect, e.g., double descent, grokking. In this study, we look at the training dynamics of the input space partition or "linear regions" formed by continuous piecewise affine DNs, e.g., networks with (leaky-)ReLU nonlinearities. First, we present a novel statistic that encompasses the *local complexity* (LC) of the DN based on the concentration of linear regions inside arbitrary dimensional neighborhoods around data points. We observe that during training, the LC around data points undergoes a number of phases, starting with a decreasing trend after initialization, followed by an ascent and ending with a final descending trend. Using exact visualization methods, we come across the perplexing observation that during the final LC descent phase of training, linear regions migrate away from training and test samples towards the decision boundary, making the DN input-output nearly linear everywhere else. We also observe that the different LC phases are closely related to the memorization and generalization performance of the DN, especially during grokking.

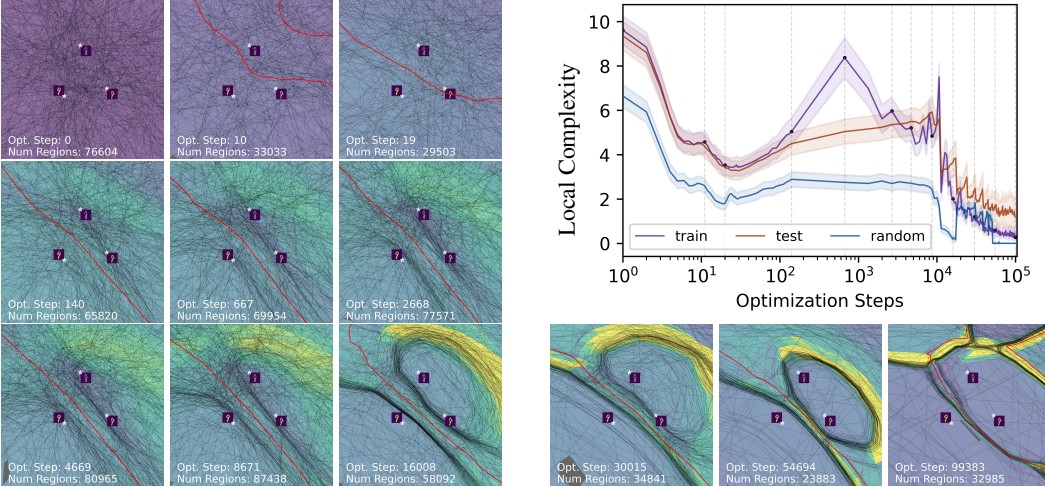

Figure 1: Evolution of the partition geometry while training a 4-layer ReLU MLP of 200 width, on $1K$ samples from MNIST. As depicted in the **top-right** figure, the local complexity around data points follow a double descent curve ultimately falling until training is stopped. *Where do the linear regions migrate to?* In the **left** and **bottom right** images we present exact visualizations of the DN partition for 2-dimensional slices of the input space crossing three training samples. We see that as training progresses, neurons (black lines) start concentrating around the training points after the first descent and move away from the training points during the second descent, ultimately concentrating near the decision boundary (red lines).

# 1 INTRODUCTION

Deep Networks (DNs) are generally known for their ability to produce state-of-the-art solutions in a wide variety of tasks LeCun et al. (2015). Beside empirical and quantitative performances, DNs have also proved revolutionary at making the machine learning community rethink some core principles such as over-parameterisation Zhang et al. (2019); Jacot et al. (2018); Bietti & Mairal (2019); Cabannes et al. (2023) and generalization Zhang et al. (2016); Balestriero et al. (2021). In fact, despite having numbers of parameters often far greater than the number of training samples, DNs rarely overfit to degenerate solutions, as opposed to other machine learning methods. Part of this comes from the many underlying impacts of the architecture Li et al. (2018); Riedi et al. (2023), the data-augmentations Hernández-García & König (2018), or even the data-ordering during mini-batch gradient descent Gunasekar et al. (2018); Barrett & Dherin (2020); Ko et al. (2023), which all provide implicit regularizers on the DN's mapping, and are not akin the usual norm-based regularizers Razin & Cohen (2020).

Among the many novel observations that came with the use of DNs, the study of its training dynamics has been among the most surprising ones exhibiting different regimes Huang & Yau (2020). For example, Swayamdipta et al. (2020) uses training dynamics for characterising datasets, Geiger et al. (2021) studies the link between training dynamics with regularization and generalization, Lewkowycz et al. (2020) relating training dynamics to the learning rate and the generalisation performance of the final solution. Most of the studies exhibit phase transitions at one point during training Papyan et al. (2020), or even demonstrate how to stretch some of the dynamics, e.g., to make generalization only emerge long after the training loss converged Power et al. (2022). The importance and implications of such studies are tremendous but all limited to the study of the loss function itself. This poses a strong limitation when the loss employed is not always perfectly aligned with the task at hand, e.g., in Self-Supervised Learning Bordes et al. (2023a;b). Another motivation to study DN's training dynamics beyond the loss function is to search for informative statistics of a DN's parameters that conveys meaningful information about its underlying mapping–a question that remains widely unanswered and underexplored to-date. As such, we pose the following question, *could we derive a statistic that is only a function of the DN's architecture and parameters, which captures its training dynamics and underlying expressivity?*

We propose a first step in that direction by proposing a novel expressivity measure of a DN that does not rely on the dataset, labels, or loss function that is used during training. Yet, the proposed measure exhibits similar dynamics as the loss function, and in fact opens new avenues to study DNs training dynamics. The proposed measure is derived from the concentration of partition regions of the DNs, in fact, our development will heavily rely on the affine spline formulation of DNs Balestriero & Baraniuk (2018). Such formulation is exact for a wide class of DNs which naturally arise form the inter-leaving of affine operators such as convolutions and fully-connected layers, and nonlinearities such as max-pooling, (leaky-)ReLU. The sole existing solution that propose to study training dynamics in such a setting is the Information Bottleneck Tishby et al. (2000) that does not follow the training loss, but nevertheless requires the training labels.

Our proposed metric is not only fast to compute for any common architecture, it also applies to most DNs out-of-the-box. We summarize our contributions below:

- We provide a novel tractable measure of a DN's local partition complexity encapsulating in a single value the expressivity of the DN's mapping; our measure is task agnostic yet informative of training dynamics

- We observe three distinct phases in training, with two descents phases and an ascent phase. We dive deeper into the phenomenon and provide for the first time a clear description of the DN's partition migrating dynamics during training; an astonishing observation showing that DN's partition regions all concentrate towards the decision boundary by the end of training

- We show that region migration can be regularized. Through a number of ablation studies we connect the training phases with over-fitting and study their changes during memorization/generalization.

We hope that our study will provide a novel lens to understand training dynamics of DNs. All the codebase for the figures and experiments will be publicly released upon completion of the review process.

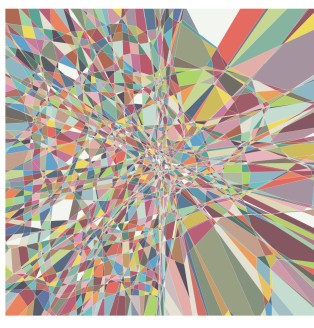 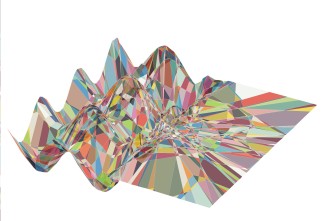

Figure 2: Visual depiction of Eq. 2 with a toy affine spline $S : \mathbb{R}^2 \rightarrow \mathbb{R}^1$. **Left.** Input space partition $\Omega$ with randomly colored regions. **Right.** graph of the affine spline function. Linear region density is significantly higher for part of the function with higher curvature.

## 2 BACKGROUND

**Deep Networks and Spline Operators.** Deep Networks (DNs) have redefined the landscape of machine learning and pattern recognition (LeCun et al., 2015). Although current DNs employ a variety of techniques that improve their performance, their core operation remains unchanged, primarily consisting of sequentially mapping an input vector $\boldsymbol{x}$ through $L$ nonlinear transformations, called *layers*, as in

$$f_\theta(\boldsymbol{x}) \triangleq \boldsymbol{W}^{(L)} \ldots \boldsymbol{a}\left(\boldsymbol{W}^{(2)} \boldsymbol{a}\left(\boldsymbol{W}^{(1)}\boldsymbol{x} + \boldsymbol{b}^{(1)}\right) + \boldsymbol{b}^{(2)}\right) \cdots + \boldsymbol{b}^{(L)}, \qquad (1)$$

starting with some input $\boldsymbol{x}$. The $\boldsymbol{W}^{(\ell)}$ weight matrix, and the $\boldsymbol{b}^{(\ell)}$ bias vector–that are collected into $\theta$–can be parameterized to control the type of layer for $\ell \in \{1, \ldots, L\}$, e.g., circulant matrix for convolutional layer. The $\boldsymbol{a}$ operator is an element-wise nonlinearity such as the Rectified Linear Unit (ReLU) (Agarap, 2018) that takes the elementwise maximum between its entry and $0$. A key result from Montufar et al. (2014); Balestriero & Baraniuk (2018) lies in tying eq. (1) to Continuous Piecewise Affine (CPA) operators. In this setting, there exists a partition $\Omega$ of the DN's input space $\mathbb{R}^D$ that is made of non-overlapping regions that overall span the entire input space –and on each such region $\omega \in \Omega$, the DN's input-output mapping is a simple affine mapping with parameters $(\boldsymbol{A}_\omega, \boldsymbol{b}_\omega)$. In short, we can express $f_\theta$ as

$$f_\theta(\boldsymbol{x}) = \sum_{\omega \in \Omega} (\boldsymbol{A}_\omega \boldsymbol{x} + \boldsymbol{b}_\omega) \mathbb{1}_{\{\boldsymbol{x} \in \omega\}}. \qquad (2)$$

Such formulations of DNNs have previously been employed to make theoretical studies amenable to actual DNNs, e.g. in generative modeling Humayun et al. (2022a;b), DNN complexity analysis Hanin & Rolnick (2019) and network pruning You et al. (2021). The spline formulation of DNNs allow leveraging the rich literature on spline theory, e.g., in approximation theory Cheney & Light (2009), optimal control Egerstedt & Martin (2009), statistics Fantuzzi et al. (2002) and related fields.

## 3 DEEP NETWORK PARTITION DENSITY

In this section we first introduce and motivate our new measure in section 3.1, we then present its implementation details in section 3.2. Following that we draw contrasts with previous methods.

### 3.1 MEASURING THE COMPLEXITY OF DEEP NETWORKS VIA PARTITION DENSITY

To make our motivation straightforward, let's first consider a supervised classification setting. By eq. (2), the learned DN's decision boundary must be linear within each region $\omega \in \Omega$. That is, the learned partition $\Omega$ must be informed about the complexity and geometry of the data distribution and task at hand to at least allow minimization of the training loss. In fact, if there exists regions $\omega \in \Omega$ containing training samples that are not linearly separable in the input space, the DN's training loss will not be minimized. Understanding the geometry of the partition $\Omega$ thus becomes as intricate and rewarding as studying the DN's mapping as a whole. With that in mind, we are now ready to introduce our measure which is a direct function of the partition $\Omega$, and nothing else, i.e., it does not require to know the labels, or even the task that the DN is being trained to solve.

Computing the Deep Network partition $\Omega$ exactly has combinatorial computational complexity. Therefore, we propose a method to approximate the density of partition regions which we coin the local DN's partition complexity. Let's consider this domain to be specified as the convex hull

of a set of vertices $\boldsymbol{V} = [\boldsymbol{v}_1, \dots \boldsymbol{v}_p]^T$ in the DN's input space. For simplicity, let's first consider a single hidden layer DN. Let's denote the DN weights $W^{(\ell)} \triangleq [\boldsymbol{w}_1^{(\ell)}, \dots, \boldsymbol{w}_{D^{(\ell)}}^{(\ell)}]$, $b^{(\ell)}$ where $\ell$ is the layer index, $D^{(\ell)}$ is the input space dimension of layer $\ell$. The partition boundary of that DN is thus formed of the hyperplanes

$$\partial\Omega = \bigcup_{d=1}^{D^{(1)}} \left\{ \boldsymbol{x} \in \mathbb{R}^{D^{(1)}} : \langle \boldsymbol{w}_i^{(1)}, \boldsymbol{x} \rangle + \boldsymbol{b}_i^{(1)} = 0 \right\}. \tag{3}$$

Moving to deep layers involve a recursive subdivision Balestriero et al. (2019) that goes beyond the scope of our study. The key insight we will leverage however is that the number of sign changes in the layer's pre-activations is a proxy for counting the number of regions in the partition. To see that, notice how input space samples who share the exact same pre-activation sign effectively lie within the same region $\omega \in \Omega$ and thus the DN will be the same affine mapping (since none of the activation functions vary). Therefore for a single layer, the local complexity for a sample in the input space can be approximated by the number of hyperplanes that intersect a given neighborhood $\boldsymbol{V}$ which is itself measure by the number of sign changes in the pre-activations that occur within that neighborhood.

## 3.2 IMPLEMENTATION

The gist of our method will be to track the number of sign changes in the layers' preactivations maps for a given neighborhood of the input space, as prescribed by the convex hull of $\boldsymbol{V}$. In particular, the method can be computed efficiently solely by forward propagation of the vertices in $\boldsymbol{V}$ as we detail below:

- Given a sample $\boldsymbol{x}$ for which we wish to compute the local complexity, we sample $P$ orthonormal vectors $\{\boldsymbol{v}_1, \dots, \boldsymbol{v}_P\}$ in the input space

- We consider as neighborhood the region defined by the convex hull, $conv(\{x \pm \boldsymbol{v}_p : p = 1, 2..., P\}) = conv(\boldsymbol{V})$

- For a given DN $f_\theta$, we take the vertices of $conv(\boldsymbol{V})$ and do a single forward pass. During the forward pass we look at the pre-activations and compute the total number units for which all the vertices do not have the same sign and sum all the layerwise values

Note that for deeper layers, our complexity measure is actually computing the number of hyperplane intersections with the convex hull of the embedded vertices instead of the convex hull of the vertices in the input space.

## 3.3 RELATION TO PRIOR WORK

The expressivity or complexity of DNs are naturally subject to the architecture, number of parameters and initialization. Montufar et al. (2014); Balestriero et al. (2019) has established precise relationship between a DN's architecture and its partition, while Raghu et al. (2017) used properties of DN partition to approximate expressivity. Humayun et al. (2023b) presented empirical observations showing that for the same DNN architecture, the partition density varies based on the complexity of the target decision boundary being learned. We compare our proposed local complexity approximation method with the local partition density approximated via SplineCAM Humayun et al. (2023b). SplineCAM is an exact computation method, therefore it has considerably more computational complexity than ours. Moreover, SplineCAM can only be used to compute the local complexity for 2D neighborhoods, whereas our method allows approximation for ND neighborhoods.

For comparison we take the depth 3 width 200 MLP that has been trained on MNIST for 100K training steps. For different training steps, we compute the local complexity via splinecam in terms of number of linear regions on a 2D plane of radius $r$, for 500 different samples. We also compute the local complexity via our proposed method for a 25-hypercube neighborhood with diagonal length $2r$. We plot the local complexity in Appendix fig. 14. We can see that for both methods the local complexity follows a similar trend with a clear double descent. Computing the local complexity for one sample takes SplineCAM significantly longer on average compared to ours, since ours is fully vectorized on GPU vs splinecam which requires CPU computations to find linear regions exactly.

## 4 LOCAL COMPLEXITY DOUBLE DESCENT

In this section we dive into the core contributions of our study, exploring the trends of the local complexity in different architectures and training regimes. The target of the experiments discussed

in this section is to understand what causes, strengthens or reduces three different phases of LC as well as the double descent behavior.

## 4.1 EXPERIMENTAL SETUP

In all our experiments, we look at the local complexity around training, test and random points sampled uniformly from the domain of the data. We perform experiments on MNIST with fully connected DNs and on CIFAR10 with CNN and Resnet architectures. For MNIST we sample the random locations for LC computation uniformly from the $[0, 1]^d$ hypercube, whereas for CIFAR10 we sample from $[-1, 1]^d$, where $d$ is the input space dimensionality. In all the MNIST experiments unless specified, we use $1K$ training samples. We use all the training samples, 10k test and 10k random samples to as locations to monitor LC. For the CIFAR10 CNN experiments, we use 2K samples, whereas in the CIFAR10 Resnet experiments we use 1K samples to compute LC. In all the results presented, we plot a confidence interval corresponding to $25\%$ of the standard deviation of LC computed. We use Purple hues to present train set LC, Orange hues to present test set LC and Blue hues to present LC at random locations.

We control the size of the local neighborhood by controlling the diagonal lengths via the $r$ parameter mentioned in section 3.2. By changing the neighborhood we look at complexity dynamics from a coarse to fine-grained viewpoint. For the MNIST and CIFAR10 experiments with fully connected and convolutional networks, we use $p = 25$ as the dimensionality of the hypercubes that define the neighborhood. For the Resnet experiments we $p = 20$. We follow the experimental setup of Liu et al. (2022) and use a Mean Square Error loss over one-hot encoded vectors for the MNIST experiments with fully connected layers. We do so to be able to draw contrasts with grokking experiments without changing the loss function. For the CIFAR10 experiments we use a cross-entropy loss. We use a learning rate of $10^{-3}$ and weight decay of $0.01$ unless specified.

## 4.2 THE ROLE OF ARCHITECTURE ON LOCAL COMPLEXITY DYNAMICS

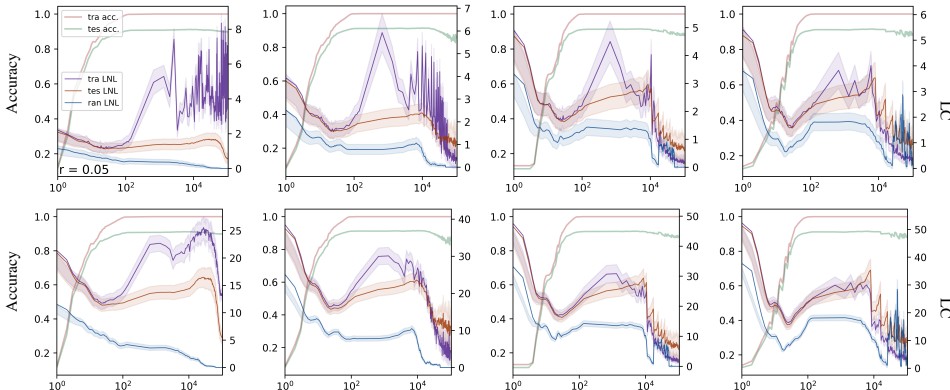

Figure 3: (From left to right) LC dynamics for MLP with depth {2,3,4,5} and width 200. Top row shows LC for a smaller neighborhood with radius $r = 0.005$, whereas bottom row shows coarse LC with $r = 0.5$. Left axis corresponds to accuracy, right axis local non linearity/local complexity. *As the number of parameters increase, the LC difference between train and test during ascent phase, decreases.*

**Depth.** In fig. 3 we plot LC during training on MNIST for Fully Connected Deep Networks with depth in $\{2, 3, 4, 5\}$ and width 200. In each plot, we show both LC as well as train-test accuracy. For all the depths, the accuracy on both the train and test sets peak during the first descent phase. During the ascent phase, we see that the train LC has a sharp ascent while the test and random LC do not.

The difference as well as the sharpness of the ascent is reduced when increasing the depth of the network. This is visible for both fine and coarse $r$ scales. For the shallowest network, we can see a second descent in the coarser scale but not in the finer $r$ scale. This indicates that for the shallow network some regions closer to the training samples are retained during later stages of training. One thing to note is that during the ascent and second descent phase, there is a clear distinction between the train and test LC. *This is indicative of membership inference fragility especially during*

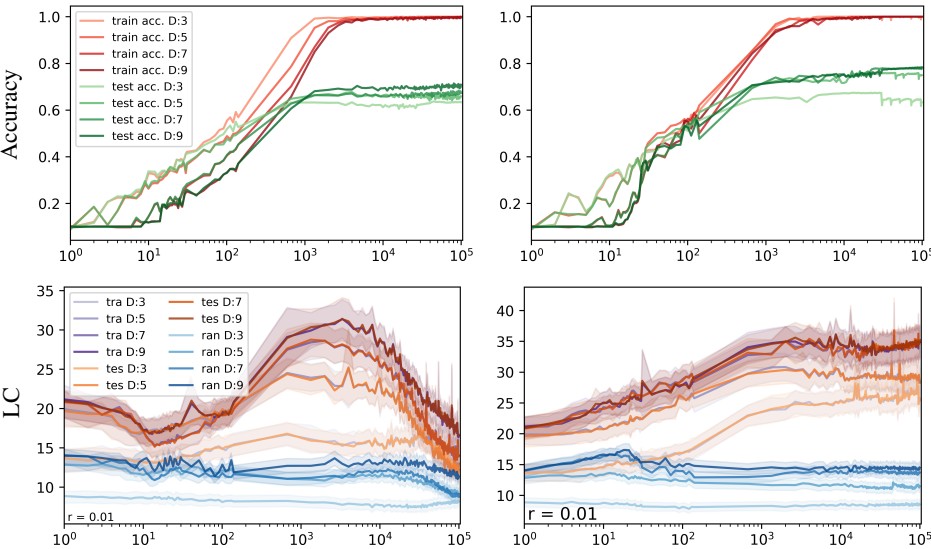

Figure 4: **Effect of Batch Normalization on the DN partition**. CNN training with varying depth on the CIFAR10 dataset with **(right)** and without **(left)** adding batch normalization after every convolutional layer. *Batch normalization removes second descent while increasing overall LC.*

*latter phases of training.* It has previously been observed in membership inference literature Tan et al. (2023), where early stopping has been used as a regularizer for membership inference. We believe the LC dynamics can shed a new light towards membership inference and the role of network complexity/capacity.

In fig. 4, we plot the local complexity during training for CNNs trained on CIFAR10 with varying depths with and without batch normalization. The CNN architecture comprises of only convolutional layers except for one fully connected layer before output. Therefore when computing LC, we only take into account the convolutional layers in the network. Contrary to the MNIST experiments, we see that in this setting, the train-test LC are almost indistinguishable throughout training. We can see that the network train and test accuracy peaks during the ascent phase and is sustained during the second descent. It can also be noticed that increasing depth increases the max LC during the ascent phase for CNNs which is contrary to what we saw for fully connected networks on MNIST. The increase of density during ascent is all over the data manifold, contrasting to just the training samples for fully connected networks.

In Appendix, we present layerwise visualization of the LC dynamics. We see that shallow layers have sharper peak during ascent phase, with distinct difference between train and test. For deeper layers however, the train vs test LC difference is negligible.

**Width.** In fig. 6 we present results for a fully connected DN with depth 3 and width $\{20, 100, 500, 1000, 2000\}$. Networks with smaller width start from a low LC at initialization compared to networks that are wider. Therefore for small width networks the initial descent becomes imperceptible. We see that as we increase width from 20 to 1000 the ascent phase starts earlier as well as reaches a higher maximum LC. However overparameterizing the network by increasing the width further to 2000, reduces the max LC during ascent, therefore reducing the crowding of neurons near training samples. *This is a possible indication of how overparameterization performs implicit regularization Kubo et al. (2019), by reducing non-linearity or local complexity concentration around training samples.*

### 4.3 EFFECT OF REGULARIZATION

Till now we have seen that overparameterization for MNIST by increasing width or depth, reduces the crowding of neurons near training samples during the ascent phase. We now look at the effect of batch normalization and weight decay on the training dynamics.

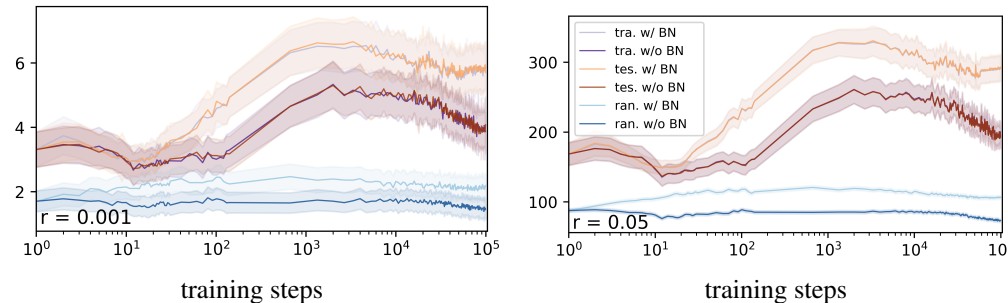

Figure 5: LC dynamics of a **Resnet18** trained with and without Batch Normalization (BN) on CI-FAR10. BN reduces second descent.

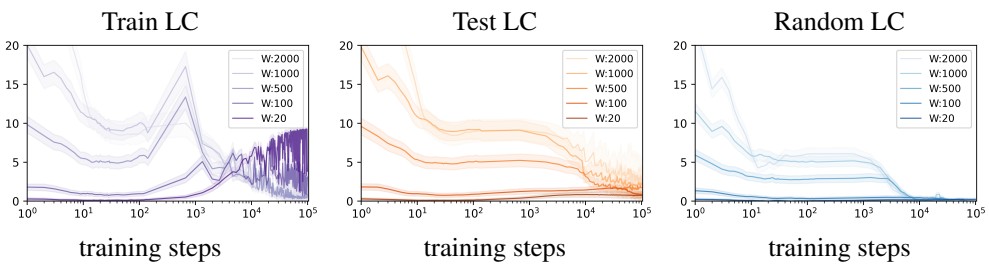

Figure 6: LC dynamics while training an MLP with **varying width** on MNIST. For the peak LC achieved around training points during the ascent phase, we see an initial increase and then decrease as the network gets overparameterized. For test and random samples, we see the LC during ascent phase saturating as we increase width.

**Batch Normalization**. It has previously been shown that Batch normalization (BN) regularizes training by dynamically updating the normalization parameters for every mini-batch, therefore increasing the noise in training Garbin et al. (2020). In fact, we recall that BN replaces the per-layer mapping from eq. (1) by centering and scaling the layer's pre-activation and adding back the learnable bias $b^{(\ell)}$. The centering and scaling statistics are computed for each mini-batch. After learning is complete, a final fixed "test time" mean $\overline{\mu}^{(\ell)}$ and standard deviation $\overline{\sigma}^{(\ell)}$ are computed using the training data. Of key interest to our observation is a result tying BN to the position in the input space of the partition region from Balestriero & Baraniuk (2022). In particular, it was proved that at each layer $\ell$ of a DN, BN explicitly adapts the partition so that the partition boundaries are as close to the training data as possible. This is confirmed by our experiments in Fig. 4 and Fig. 5 where we present results for CNN and Resnet trained on CIFAR10, with and without BN. The first thing to note is that BN largely removes the second descent for all the experiments. This is intuitive since that second descent, as also witness in fig. 1, emerges from the partition regions migrating towards the decision boundary which is further away from the training. As a result, while the training dynamics may aim for a second descent, BN ensure that at each layer the hyperplanes from eq. (3) remain close to the training mini-batch. This is further confirmed by the second observation that employing BN increases the overall LC throughout training for train, test and also random points in the input space, i.e. the DN's partition does not concentrate in a specific region of the space.

**Weight Decay** regularizes a neural network by reducing the norm of the network weights, therefore reducing the per region slope norm as well. We train a CNN with depth 5 and width 32 and varying weight decay. In Fig. 7 we present the train and random LC for our experiments. We can see that increasing weight decay also delays or removes the second descent in training LC. Moreover, strong weight decay also reduces the duration of ascent phase, as well as reduces the peak LC during ascent. This is dissimilar from BN, which removes the second descent but increases LC overall.

## 4.4 MEMORIZATION AND GENERALIZATION

In this section we present a number of experiments through which we try to understand the memorization and generalization characteristics of DNs during the three different phases of training.

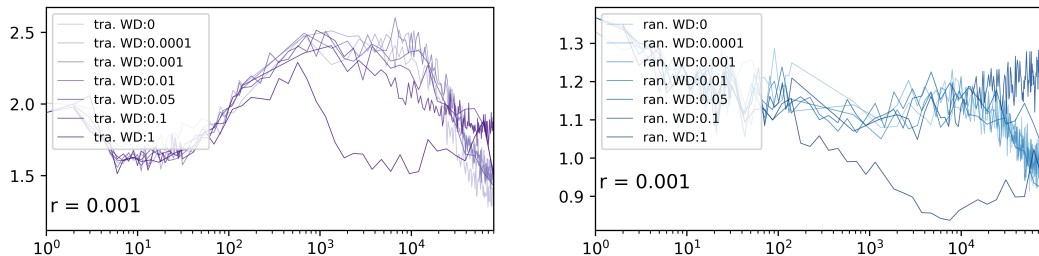

Figure 7: Effect of **weight decay** on the LC dynamics of a CNN trained on CIFAR10. Final descent gets reduced as we increase weight decay.

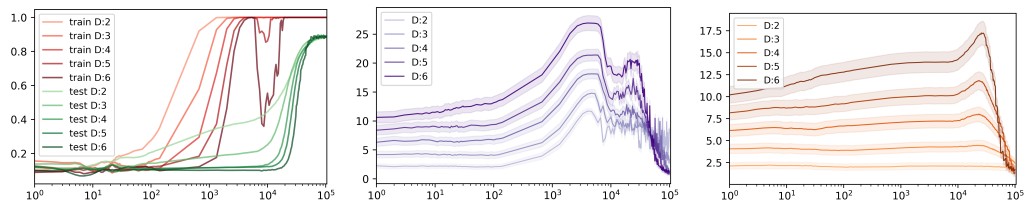

Figure 8: Accuracy, train and test LC dynamics for an MLP trained on MNIST when **grokking** is induced. *As depth is increased the max LC during ascent phase becomes larger. We can also see a distinct second peak right before the descent phase.*

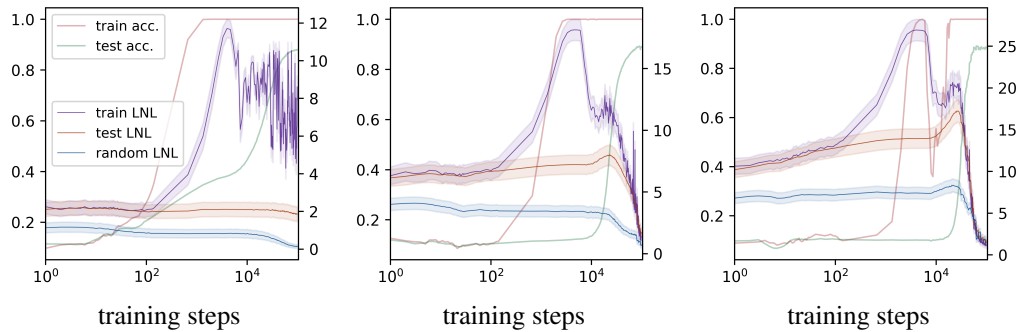

Figure 9: Effect of depth on grokking and local complexity dynamics. Memorization phase of grokking happens during the ascent phase of LC. Generalization occurs during the second descent of LC.

**Grokking.** First we look at Grokking, as a training setup to disconnect memorization from generalization. Grokking Power et al. (2022) is a phenomenon where networks tend to generalize significantly after overfitting, on small algorithmic task such as modular addition. Liu et al. (2022) showed that grokking can occur in non-algorithmic datasets as well, e.g., while training MNIST with a fully connected network. Following Liu et al. (2022), we induce grokking by scaling the weights of a fully connected DN by 8 at initialization. In fig. 8 we present the LC dynamics when grokking is induced for networks with varying depth. Note that for shallow fully connected networks, generalization starts earlier than deeper DNs. Here, increasing the depth of the network makes the ascent max LC peak higher, contrary to the general discussed in section 4.2, where increasing depth reduced the max LC peak as well as the difference between train and test LC during ascent. We also see that increasing depth delays generalization, as well as makes LC drop faster during the second descent phase. The first descent phase is almost non-existent with ascent happening from early training. *This indicates that the ascent phase during grokking, is related to memorization.* Increasing the depth, increases the capacity of the model to memorize Arpit et al. (2017), therefore increasing the ascent peak.

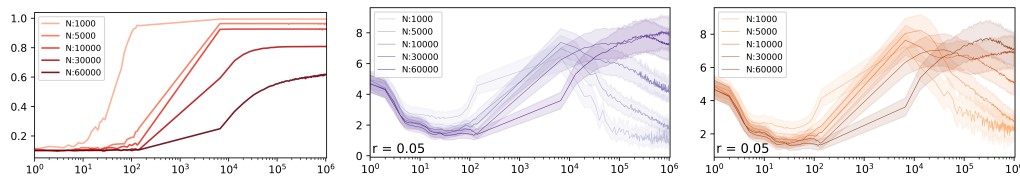

Figure 10: LC dynamics of a depth 3 width 200 MLP with **varying number of MNIST training samples**. As data set size is increased, the network generalizes more and earlier. We see that with more generalization removes the LC peak during ascent phase.

Figure 11: LC dynamics of a depth 3 width 200 MLP with **varying number of MNIST training samples with random labels**. Here more training samples would require more memorization, therefore we do not see the ascent peak get reduced, like we see with non-random labels.

In the grokking setting, we can prominently see a second peak or a second ascent happening right after the first ascent. In fig. 9 we can see that the second ascent occurs right before the model starts 'grokking', i.e., delayed generalization. *The presence of two ascents during grokking is possible indication that the network increases the local complexity near training points both during memorization and generalization.*

**Effect of Training Data.** In the following set of experiments, we control the training dataset to either induce higher generalization on higher memorization. Recall that in our MNIST experiments, we use $1k$ training samples. Increasing the number of samples would therefore increase the generalization of our network. On the other hand we also sweep the dataset size for a random label memorization task. In this setting, higher dataset size would require more memorization, therefore more capacity.

In fig. 10 and fig. 11, we present train and test LC, along with training accuracy curves. We vary the size of training dataset without and with randomized labels respectively. We see that increasing the dataset size for ground truth labels, increases generalization, which gradually removes the ascent phase. The early ascent during grokking and removed ascent during early generalization further strengthens the connection between the ascent phase and overfitting.

Training on a set of randomly labeled samples is a task that requires memorization or overfitting on training data. Therefore, gradually increasing training dataset size elongates the ascent phase, gradually increasing the peak train and test LC.

## 5   CONCLUSION

We have provided the first empirical study of the training dynamics of Deep Networks (DNs) partitions and brought forward a surprising observation where the number of partition region first grows around training samples, to then reduce up until generalization starts emerging, at which point the density increases again to finally decrease up until end of training. Crucially, we were able to pinpoint the migration phenomenon of the DN's partition to concentrating around the decision boundary, and thus far away from the training samples. This observation is the first to notice how the partition evolve so dynamically during training. Our finding not only rebutted some pre-existing studies that relied on incorrect stationary assumption of the partition, but also demonstrate that some of the core dynamics of DN training remain not only unexplained, but actually uncovered.

## ACKNOWLEDGEMENTS

Humayun and Baraniuk were supported by NSF grants CCF1911094, IIS-1838177, and IIS-1730574; ONR grants N00014- 18-12571, N00014-20-1-2534, and MURI N00014-20-1-2787; AFOSR grant FA9550-22-1-0060; and a Vannevar Bush Faculty Fellowship, ONR grant N00014-18-1-2047.

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

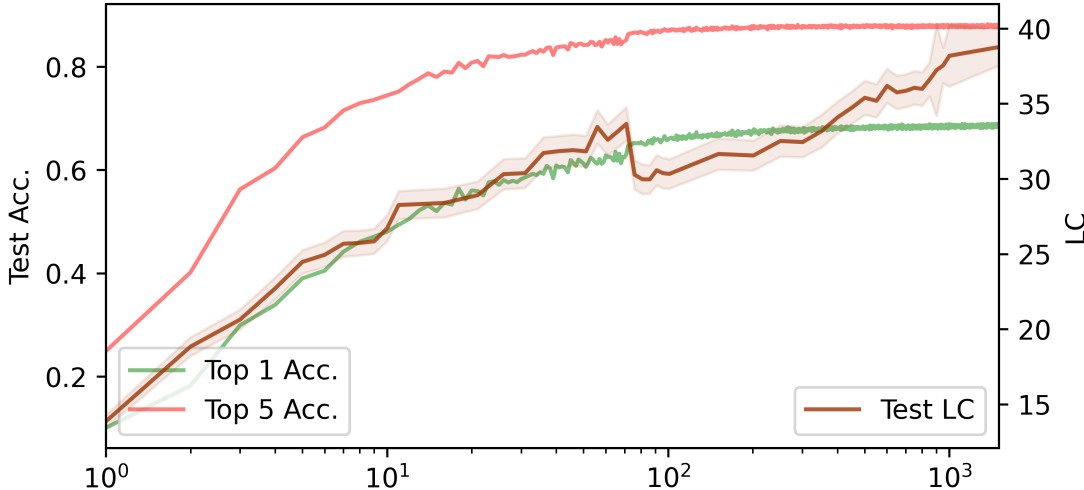

Figure 12: LC during Imagenet training of a Resnet18. LC is computed only on test points using 1000 test set samples. Computing LC 1000 samples takes approx. 28s on an RTX 8000.

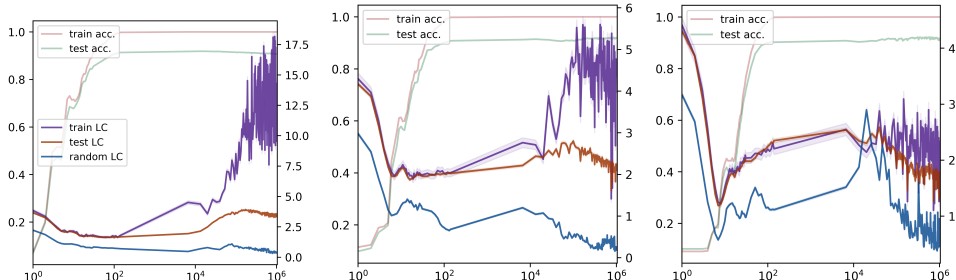

Figure 13: LC dynamics for a GeLU-MLP with width 200 and depth $\{3, 4, 5\}$ presented from left to right. LC is calculated at 1000 training points and 10000 test and random points during training on MNIST. For all of the settings we see double descent occuring in the LC dynamics.

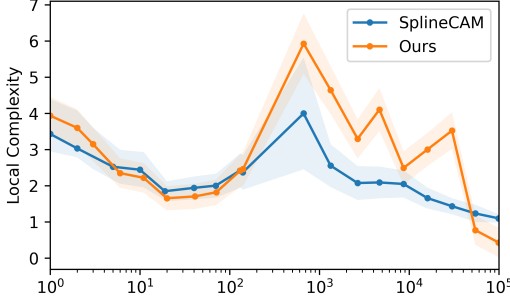

Figure 14: Comparing the local complexity measured in terms of the number of linear regions computed exactly by SplineCAM Humayun et al. (2023b) and number of hyperplane cuts by our proposed method. Both methods exhibit the double descent behavior.

## A    APPENDIX

## B    EXTRA FIGURES

## C    EMPIRICAL ANALYSIS OF OUR PROPOSED METHOD

Computing the exact number of linear regions or piecewise-linear hyperplane intersections for an deep network with N-dimensional input space neighborhood has combinatorial complexity and therefore is intractable. This is one of the key motivations behind our approximation method.

**MLP with zero bias.** To validate our method, we start with a toy experiment with a linear MLP with width 400, depth 50, 784 dimensional input space, initialized with zero bias and random weights. In such a setting all the layerwise hyperplanes intersect the origin at their input space. We compute the LC around the input space origin using our method, for neighborhoods of varying radius $r = \{0.0001, 0.001, 0.01, 0.1, 1, 10\}$ and dimensionality $P = \{2, 10, 25, 50, 100, 200\}$. For all the trials, our method recovers all the layerwise hyperplane intersections, even with a neighborhood dimensionality of $P = 2$.

**Non-Zero Bias Random MLP with shifting neighborhood.** For a randomly initialized MLP, we expect to see lower local complexity as we move away from the origin Hanin & Rolnick (2019). For this experiment we take a width 100 depth 18 MLP with input dimensionality $d = 784$, Leaky-ReLU activation with negative slope 0.01. We start by computing LC at the origin $[0]^d$, and linearly shift towards the vector $[10]^d$. We see that for all the settings, shifting away from the origin reduces LC. LC gets saturated with increasing $P$, showing that lower dimensional neighborhoods can be good enough for approximating LC. Increasing $r$ on the other hand, increases LC and reduces LC variations between shifts, since the neighborhood becomes larger and LC becomes less local.

**Trained MLP comparison with SplineCam.** For non-linear MLPs, we compare with the exact computation method Splinecam Humayun et al. (2023a). We take a depth 3 width 200 MLP and train it on MNIST for 100K training steps. For 20 different training checkpoints, we compute the local complexity in terms of the number of linear regions computed via SplineCam and number of hyperplane intersections via our proposed method. We compute the local complexity for 500 different training samples. For both our method and SplineCam we consider a radius of 0.001. For our method, we consider a neighborhood with dimensionality $P = 25$. We present the LC trajectories in Fig. 14. We can see that for both methods the local complexity follows a similar trend with a double descent behavior.

**Deformation of neighborhood by deep networks.** As mentioned in Sec.3.2, we compute the local complexity in a layerwise fashion by embedding a neighborhood $conv(V)$ into the input space for any layer and computing the number of hyperplane intersections with $conv(V^\ell)$, where $V^\ell$ is the embedded vertices at the input space of layer $\ell$. The approximation of local complexity is therefore

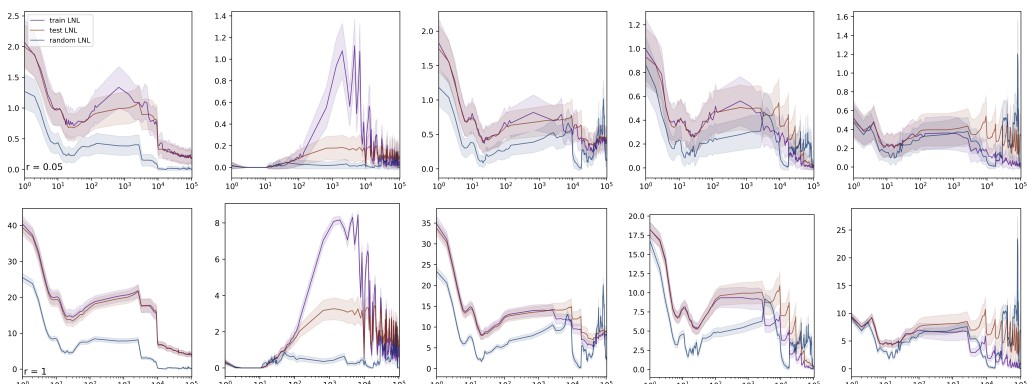

Figure 15: Layerwise LC dynamics from shallow to deeper layers **(left to right)** for a depth 6 and weight 200 MLP. First and second row corresponds to lower and higher $r$. Shallow layers have sharper peak during ascent phase, with distinct difference between train and test. For deeper layers however, the train vs test LC difference is negligible.

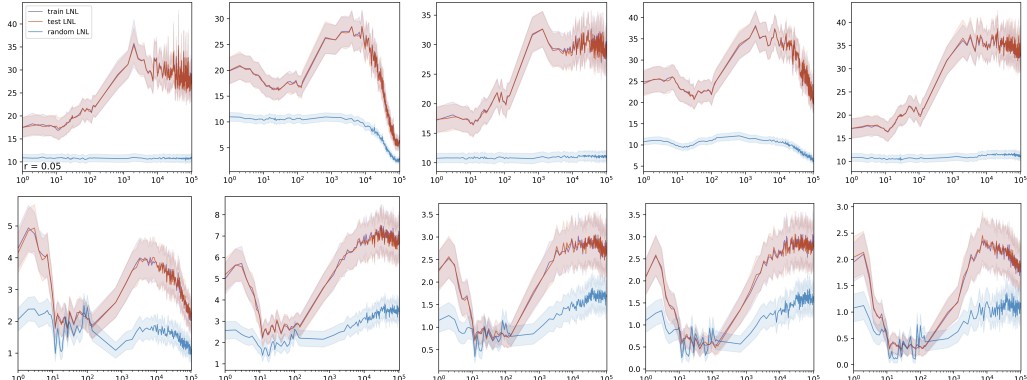

Figure 16: LC dynamics for the (Top Row) shallowest and deepest (Bottom Row) layers of a Resnet18 without BN, while training on CIFAR10. Notice how for the shallow layers, the presence of the second descent alternates between layers. We see that for the first ReLU of shallow pre-act resblocks, the second descent does not occur while it occurs for the second ReLU. For deeper layers we see that before the ascent phase, there exists a period when the train and test LC matches the random LC.

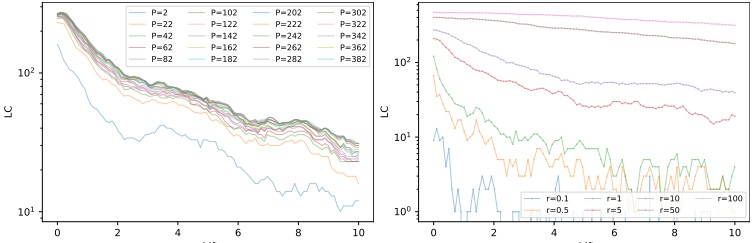

Figure 17: LC for a $P$ dimensional neighborhood with radius $r$ while being shifted from the origin $[0]^d$ to vector $[10]^d$. In **left**, we vary $P$ with fixed $r = 5$ while on **right** we vary $r$ for fixed $P = 20$. We see that for all the settings, shifting away from the origin reduces LC. The increase of LC with the neighborhood dimensionality $P$ gets saturated as we increase $P$, showing that lower dimensional neighborhoods can be good enough for approximating LC. Increasing $r$ on the other hand, increases LC and reduces LC variations between shifts, since the neighborhood becomes larger and LC becomes less local.

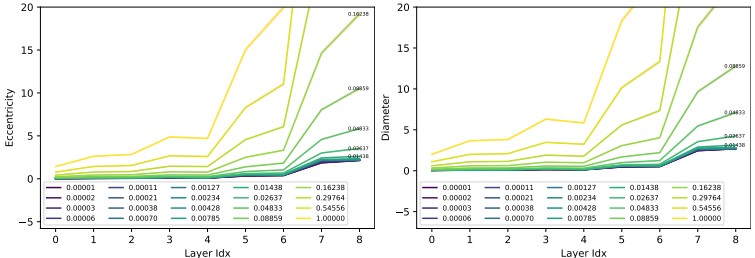

Figure 18: Change of avg. eccentricity and diameter Xu et al. (2021) of the input space neighborhood by different layers of a CNN trained on the CIFAR10 dataset. For different sampling radius $r$ of the sampled input space neighborhood $V$, the change of eccentricity and diameter denotes how much deformation the neighborhood undergoes between layers. Here, layer 0 corresponds to the input space neighborhood. Numbers are averaged over neighborhoods sampled for 1000 training points from CIFAR10. For larger radius the deformation increases with depth exponentially. For $r \leq 0.014$ deformation is lower, indicating that smaller radius neighborhoods are reliable for LC computation on deeper networks. Confidence interval shown in red, is almost imperceptible.

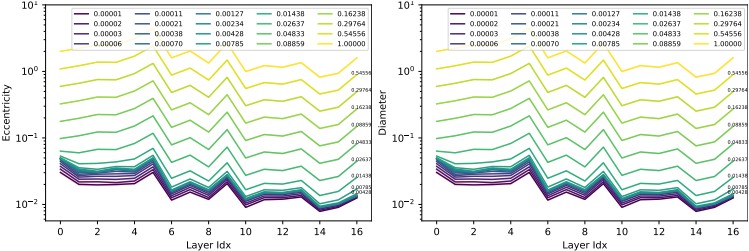

Figure 19: Change of avg. eccentricity and diameter Xu et al. (2021) of the input space neighborhood by different layers of a Resnet18 trained on the CIFAR10 dataset, similar to the setting of Fig. 18. Resnet deforms the input neighborhood by reducing the avg. eccentricity and diameter of the neighborhood graphs. For $r \leq 0.014$ deformation is lower, indicating that smaller radius neighborhoods are reliable for LC computation on deeper networks.

subject to the deformation induced by each layer to $conv(V)$. To measure deformation by layers 1 to $\ell-1$, we consider the undirected graph formed by the vertices $V^\ell$ and compute the average eccentricity and diameter of the graphs Xu et al. (2021). Eccentricity for any vertex $v$ of a graph, is denoted by the maximum shortest path distance between $v$ and all the connected vertices in the graph. The diameter is the maximum eccentricity over vertices of a graph. Recall from Sec.3.2 that $conv(V)$ where $V = \{x \pm rv_p : p = 1...P\}$ for an input space point $x$, is a cross-polytope of dimensionality $P$, where only two vertices are sampled from any of the orthogonal directions $v_p$. Therefore, all vertices share edges with each other except for pairs $\{(x + rv_p, x - rv_p) : p = 1...P\}$. Given such connectivity, we compute the average eccentricity and diameter of neighborhoods $conv(V^\ell)$ around 1000 training points from CIFAR10 for a trained CNN (Fig. 18). We see that for larger $r$ both of the deformation metrics exponentially increase, where as for $r \leq 0.014$ the deformation is lower and more stable. This shows that for lower $r$ our LC approximation for deeper CNN networks would be better since the neighborhood does not get deformed significantly.

