# OpenReview forum: "Deep Network Partition Density Exhibits Double Descent"
_ICLR.cc/2024/Conference — Submitted to ICLR 2024_

### Official Review · Reviewer_dy7R · 2023-10-29

**Soundness:** 3 good
**Presentation:** 3 good
**Contribution:** 3 good
**Rating:** 6
**Confidence:** 3

**Summary:**

This paper proposes a new measure of the local complexity of functions that deep networks (abbrv. DN) learn throughout training.  The authors develop this measure as a follow up to work establishing a large class of DNs as a composition of max-affine spline operators, which distinguishes their model of DNs from those that uses information theoretic tools.  They use their measure of complexity to examine different phenomena exhibited by DNs during training (grokking, double descent, memorization), and explain them in light of their local complexity formalism.

**Strengths:**

- This paper is well written, and builds solidly on prior work by [Balestriero et al.](https://proceedings.mlr.press/v80/balestriero18b/balestriero18b.pdf).
- The motivation of the theory is well developed in section 3.1
- The experiments are well motivated also, and address some of the most relevant phenomena observed in DN training dynamics.
- Many other papers that attempt to explore local representations of deep network points usually rely on a trust region about a point, and make assumptions about the geometry of a latent representation space.  It's interesting to me that the authors instead use a convex hull to construct the neighbourhood, though they might want to spend a bit more time motivating this choice (if it's the best choice given their [earlier work](https://proceedings.mlr.press/v80/balestriero18b/balestriero18b.pdf)., another citation would not go amiss here).

**Weaknesses:**

- The biggest weakness in the paper seems to be that a critical part of it is missing.  The experimental setup in section 4.1 states
> We control the size of the local neighborhood by controlling the diagonal lengths via the $r$ parameter mentioned in section 3.2

But section 3.2 makes no mention of a parameter $r$, and it is difficult to make a full evaluation of the experiments in section 4 absent a definition of $r$.

- There is a line of prior work that seems closely related to the authors' own proposed work, which is the idea of polytope lenses by [Black et al.](https://arxiv.org/abs/2211.12312).  Briefly, this work attempts to look at polysemy of neurons by using ideas from casting neural networks as compositions of splines, though they are chiefly interested in interpretation rather than complexity.  I think this paper would be made stronger by contrasting the ideas in Black et al to this work's own, by showing the benefit of viewing DNs as splines has multiple uses.

- The lack of a need for labels makes this measure ripe for helping to help explain the effects of self-supervised learning.  I'm a bit surprised that the authors don't include experiments for why (e.g) BYOL can work by measuring the local complexity of teacher & student networks as they train, and iteratively replace each other.
- This is a minor point, but it would be helpful for the authors to spend a few words about why the cost of computing the $P$ orthonormal basis vectors for each $x$ either isn't prohibitive, or why it might be in certain cases.
- Re-reading section 3.1, I'm surprised to see no characterization of the properties of $[\mathbf{v}_1, \dots, \mathbf{v}_P]$ other than they form an orthonormal set, and that $x$ is within $conv(\mathbf{V})$.  This latter point is implied but not stated.  Maybe 3.1 could do with some more care in specifying the relation between $\mathbf{x}$ and $\mathbf{V}$?

Other minor points:
- italicized sentence at the end of the second paragraph of the introduction has some typos.  But it's compelling!
-  If this sampling procedure is for each sample input, it’s going to become really expensive computationally.  How do they compute a succession of P orthonormal vectors, is this by Gram-Schmidt (or similar procedure?). Is there any way to amortize this complexity?  I think it would help make this measure more practically helpful

**Questions:**

- How stable is local complexity about a point $\mathbf{x}$ as a consequence of the choice of $\mathbf{V}$?  And choice of $\mathbf{P}"?  These seem like important questions to answer.
- The authors intentionally leave out extension of this to deeper networks, as a clear continuation of this line of work.  But I think the conclusion would be strengthened by spending some more of the word budget to help lead the reader into how the authors plan to do so.
- Do the authors have any conjecture about how the partition migrating dynamics relate to the phenomenon of neural collapse?

To conclude, I think this is a good paper that needs a bit more work and polishing to become a really good paper.  If the authors will consider the points I've raised here (and in the weaknesses section), I'm willing to raise my score.

---

> ### Author Response · Authors · 2023-11-23
> **Author Response Part 1**
>
> We thank the reviewer for their positive comments, especially their appreciation of the motivation behind our work.
>
> >  The biggest weakness in the paper seems to be that a critical part of it is missing. The experimental setup in section 4.1 states
>
> We thank the reviewer for raising this very important point. We have re-written section 4.1 and 3.1 adding the definition of the radius $r$ and its relationship with $conv(V)$. In the appendix, we have also added experiments showing the effect of varying $r$ and number of orthonormal vectors $P$.
>
> To summarize, we start by sampling $P$ orthonormal vectors $B=${$ v_p : p=1...P$} and use them to denote vertices of a local neighborhood $V$ around any data point $x$ s.t. $V=${$x \pm rv_p : p = 1…P$} where $r$ is the distance from $x$ at which we choose vertices defining the neighborhood. Therefore, $x$ is the centroid of $V$ consisting of 2P vertices. We define the convex hull $conv(V)$ as the local neighborhood of $x$ in the input space. The dimensionality and volume of the neighborhood can be controlled via $P$ and $r$. Each neighborhood in the input space around any data point $x$ is a cross-polytope or the dual of a hypercube with dimensionality P and diagonal length $2r$. Each neighborhood can also be considered a unit $\ell_1$-norm ball scaled by radius $r$.
>
> > There is a line of prior work that seems closely related to the authors' own proposed work, which is the idea of polytope lenses by Black et al.
>
> We thank the reviewer for bringing the paper to our attention. Indeed the analysis by Black et al. is very relevant to the results of our paper. While they are focusing on interpretability, the authors provide experiments showing that the density of polytope boundaries are related to semantic boundaries. In our paper, while we compute local complexity in terms of layerwise hyperplane intersections for a given neighborhood, the number of polytope edges formed via an arrangement of $N$ different $(d-1)$-hyperplanes is of the order $N^d$. Therefore, our proposed local complexity would be highly correlated with polytope boundary density. We will add a short discussion contrasting our method with Black et al. in our conclusion and connect it with potential future directions on interpretability.
>
> >  I'm a bit surprised that the authors don't include experiments for why (e.g) BYOL can work by ...
>
> We thank the reviewer for raising this excellent point. Indeed, evaluating BYOL especially the alignment between the teacher and student network, can be assessed in a sample-wise manner using our local complexity metric. In this paper however, we have mostly focused on the double descent phenomenon of local complexity that we observe in a wide variety of settings, while providing a method to do fast approximation of local complexity. The double descent phenomenon is important since it shows that based on the phase of training, neural networks tend to increase or decrease local complexity around training points, which opens up avenues for new theoretical studies on neural network training and generalization. We leave the SSL experiments for future work and will add a note on that in the conclusion.
>
> > ...about why the cost of computing the orthonormal basis vectors for each either isn't prohibitive...
>
> > If this sampling procedure is for each sample input, it’s going to become really expensive computationally.
>
> In our experiments, we have seen no significant difference between sampling the orthonormal basis vectors for every data point $x$, vs using the same orthonormal frame being used for all the data points. Using the Pytorch implementation of sampling P orthonormal vectors, finding P=25 vectors takes approximately 9.49 ms, which needs to be done only once at the beginning of our experiments. We have added these details in the paper.
>
> > Re-reading section 3.1, I'm surprised to see no characterization of the properties of
>
> We apologize for the lack of clarity, we are rewriting Sec 3.1 and clarifying the notations as discussed above.

---

> > ### Author Response · Authors · 2023-11-23
> > **Author Response Part 2**
> >
> > > How stable is local complexity about a point $x$ as a consequence of the choice of $V$?
> >
> > In appendix, we have added experiments exploring how the parameters of neighborhood $conv(V)$, namely radius $r$ and dimensionality $P$ affects LC computation. In summary, we provide the following observations:
> >
> > 1. As we increase P, after a certain point, there is marginal increase in LC with increasing P for the same input space point $x$. In Fig.16 left we show that for a 784 dimensional input space and a randomly initialized MLP with width 100 and depth 18, $P=42$ has marginal difference in LC compared to $P=382$.
> >
> > 2. As we increase radius $r$, the volume of the neighborhood $conv(V)$ increases. While higher neighborhood volume allows computing LC at coarser scales (Fig.16 right), it can also introduce distortion in the embedded neighborhood reducing approximation quality. In this regard, we compute the eccentricity and diameter of the graph formed via $V^\ell$ which is the layer $\ell-1$ embeddings of vertices $V$.  We use $1000$ training points from CIFAR10 to define the neighborhoods for a trained CNN. We see that for larger $r$ both of the deformation metrics exponentially increase, whereas for $r \leq 0.014$ the deformation is lower and more stable. This shows that for lower $r$ our LC approximation for deeper CNN networks is stable since the neighborhood does not get deformed significantly.
> >
> > > extension of this to deeper networks
> >
> > In appendix, we have added an experiment computing the LC while training a Resnet18 on Imagenet upto 69% top-1 accuracy. Computing LC for 1000 imagenet samples require ~28s on a Quadro RTX 8000 GPU. We use a hull dimensionality of $P=25$ and radius $r=0.001$. We have also previously presented, local complexity dynamics for Resnet18 trained on CIFAR10, and shown that without Batch Normalization, it exhibits similar dynamics as we have seen in CNNs and MLPs.
> >
> > > Do the authors have any conjecture about how the partition migrating dynamics relate to the phenomenon of neural collapse?
> >
> > We thank the reviewer for this excellent point. In Fig.1, we can see that as training progresses, the neurons (denoted by black lines) start orienting themselves with each other and grouping together. We call this phenomenon partition migration. This can be further seen in the animation we have provided as supplementary material, which presents an exact visualization of the dynamics presented in Fig.1 as training progresses. The grouping of neurons is equivalent to the neurons losing linear independence, which is directly related to neural collapse. There are two additional observations in our result that to the best of our knowledge have not been explored previously in Neural Collapse literature. The first, neurons from different layers of the network may ‘collapse’ together, meaning that neural collapse is not a layerwise phenomenon. Secondly, as can be seen in the supplementary animation, the grouped neurons can jump between different orientations while staying grouped together, indicating that a network may keep changing its features locally while in a collapsed form. We will add further discussions on this point taking into account space limitations.

---

### Official Review · Reviewer_prSj · 2023-10-29

**Soundness:** 2 fair
**Presentation:** 2 fair
**Contribution:** 2 fair
**Rating:** 3
**Confidence:** 3

**Summary:**

The authors propose an approximation for computing continuous piecewise affine operators in deep (ReLU, I believe) networks and then study how the (approximate) partitions evolve during model overtraining.

**Strengths:**

Overall, I think this paper is well written.

**Weaknesses:**

Overall, I think that this paper has a number of significant shortcomings. To summarize, I think

1. the motivation is weak
2. the authors dodge the hard central research question of how to make the continuous piecewise affine (CPA) operator perspective useful at scale
3. the method is not well explained and is based on an approximation that, to the best of my ability to discern, goes untested
3. the experiments are too simple (e.g, 1k samples from MNIST) or odd (e.g., training MNIST for 100K training steps, which is massive overkill, or changing how the MNIST targets are encoded to induce grokking)
4. the experimental results are messy and at times unclear, with dubious connection to double descent

In detail:

## Motivation

> No other statistics about the DN has been found to be as informative as the loss func-
tion

I think this claim is untrue. My understanding is that there is work in the deep kernel literature in both lazy learning and feature learning regimes, e.g., [1, 2] and mechanistic interpretability literature, e.g., [3, 4] about studying DN learning dynamics using more granular statistics than the loss function.

If the authors do wish to advocate for a CPA perspective, which is valid, they should make a case of (1) what properties we want deep network learning statistics to reveal, (2) why other existing approaches are better and (3) why CPA is the right way to go. But I don't see this, and thus I am left wondering "Sure, we _can_ take a CPA approach, but what are we looking for that we don't already have and that CPA can provide?"

[1] Bordelon, Canatar & Pehlevan ICML 2022. Spectrum Dependent Learning Curves in Kernel Regression and Wide Neural Networks.

[2] Bordelon & Pehlevan NeurIPS 2022. Self-Consistent Dynamical Field Theory of Kernel Evolution in Wide Neural Networks.

[3] Nanda, et al. ICLR 2023. Progress measures for grokking via mechanistic interpretability.

[4] Liu, Kitouni, Nolte, Michaud, Tegmark, Williams. NeurIPS 2022. Towards Understanding Grokking: An Effective Theory of Representation Learning

> In this study, we provide a novel statistic that measures the underlying DN’s
local complexity, exhibiting two key benefits: [...] (ii) it is informative about the training loss and accuracy dynamics.

Previous work gave loss error bounds as a function of the number of non-overlapping regions in continuous piecewise affine operators in DN [1] and studied their performance empirically, so this manuscript’s second main claim (that changing local complexity results in changing loss and changing accuracy) seems unsurprising.

At a minimum, I would recommend the authors cite [1]. I also happened to attend a talk by the first author of [1] and Professor Shai Ben-David in the audience said that the authors of [1] were ignoring significant amounts of prior work from the 80s-2000s; while I myself don’t know the correct citations, I suspect that the authors of this manuscript might also want to do an older literature search. (Note: I am unaffiliated with [1] and with Professor Ben-David.)

I should also note that this manuscript’s Fig 2 almost exactly matches [1]’s Fig 1b.

[1] Ji, Pascanu, Hjelm, Lakshminarayanan, Vedaldi CoLLAs 2022. Test Sample Accuracy Scales with Training Sample Density in Neural Networks.

> In fact, our development will heavily rely on the affine spline formulation of DNs
Balestriero & Baraniuk (2018). Such formulation is exact for a wide class of DNs which naturally
arise [...]  nonlinearities such as max-pooling, (leaky-)ReLU.

I might be mistaken, but does the CPA perspective apply to more modern nonlinearities? Balestriero and Baraniuk 2018 studied ReLU, Leaky ReLu and absolute value, but many modern models use newer nonlinearities like GeLU and SwiGLU, used in ViT and LLMs e.g. Llama 2. I guess my question is: to what extent is CPA applicable only to certain classes of nonlinearities?

## Method

> Moving to deep layers involve a recursive subdivision Balestriero et al. (2019) that goes beyond the scope of our study.

My understanding is the key reason why the CPA perspective has struggled is because of the combinatorial complexity. Waving this aside amounts to waving aside the key research challenge with CPAs.

> Equation 3

What is the notation $\partial \Omega$? Is equation 3 a definition?

> The key insight we will leverage however is that the number of sign changes in the layer’s pre-activations is a proxy for counting the number of regions in the partition. To see that, notice how input space samples who share the exact same pre-activation sign effectively lie within
the same region ω ∈ Ω and thus the DN will be the same affine mapping (since none of the activation functions vary). Therefore for a single layer, the local complexity for a sample in the input space can be approximated by the number of hyperplanes that intersect a given neighborhood V which is itself measure by the number of sign changes in the pre-activations that occur within that neighborhood.

I think this is a key point that should be explained in more detail, with examples and/or figures and/or equations.

Also, doubling back to my question, I’m unsure of whether the insight still holds with depth. One would imagine that multiple regions at layer $i$ might be mapped to the same region at layer $i+1$ or $i+2$.

> Note that for deeper layers, our complexity measure is actually computing the number of hyperplane intersections with the convex hull of the embedded vertices instead of the convex hull of the vertices in the input space.

This comment is somewhat concerning. This comment suggests that the quantity one desires (the convex hull of the vertices in input space) has subtly shifted to a different quantity (the convex hull of the embedded vertices) and how these two are related (if at all) is not analyzed, discussed or explored mathematically or empirically.

> We control the size of the local neighborhood by controlling the diagonal lengths via the r parameter mentioned in section 3.2.

I cannot identify an $r$ parameter in Section 3.2, and this $r$ parameter seems to be important for subsequent experiments.

## Experiments

Before delving into the experiment, the proposed approximation (Section 3.1 & 3.2) is not yet validated. One would first want to know: how well does this method approximate the full (and extremely expensive) partition computation? How many orthogonal vectors are necessary to get a good approximation, and how does this depend on the data, architecture, and other implementation choices? What effect does the parameter $r$ have? As best as I can tell, there is no answer to these questions.

The closest answer I can find is Section 3.3, which compares this method against SplineCAM in 1 architecture (depth 3 width 200 MLP) on 500 MNIST samples. This feels inadequate to me for two reasons:
1. I want a comparison to an exact baseline. But whether SplineCAM is exact is unclear. The authors call SplineCAM both an approximation (“with the local partition density approximated via SplineCAM”) and exact (“SplineCAM is an exact computation method”). SplineCAM also appears to be limited (“SplineCAM can only be used to compute the local complexity for 2D neighborhoods”), which an exact method should not be.
2. Even if SplineCAM is exact, I see no characterization of how the number of orthogonal dimensions, or $r$, or architecture, or dataset plays a role.

Consequently, I am left with no understanding of how good this approximation is or under what conditions the approximation is good. This makes me skeptical of the subsequent experimental results.

> We perform experiments on MNIST with fully connected DNs and on CIFAR10 with CNN and Resnet architectures

Despite this claim, it appears to me that _most) of the experiments focus on a subset of 1k data from MNIST (Figures 1, 3, 6, 7, 8, 9, 10). The CIFAR experiments, while fewer and less prominent, are performed on 2.5k data. In 2023, such small networks trained on such simple data feels too inadequate. Moreover, the training diets seem bizarre. Most of the models appear to display high plateau accuracy within ~100 optimization optimization steps, but training continues till 100000 training steps without motivation or explanation. Models are known to exhibit strange behaviors with such long training diets (e.g., neural collapse) so I am skeptical that we should be studying networks in such late atypical stages. I personally have never seen anyone train MNIST to 100k gradient steps unless they're looking for atypical model behavior.

> For the MNIST and CIFAR10 experiments with fully connected and convolutional networks, we use p = 25 as the dimensionality of the hypercubes that define the neighborhood.

I don’t know why p=25 was chosen, and it would be good to see some sensitivity analysis to how much $p$ affects the results.

> As depicted in the top-right figure, the amount of region near training samples follow a double descent curve ultimately falling until training is stopped

There are two oddities with the Figure 1 result.

1. Double descent is almost always observed as a function of parameters to data, not optimization steps.
2.  The _train_ loss peaks, even though double descent is characterized by the _test_ loss peaking.

Now, perhaps the authors do not mean “double descent” in the way that is commonly used. If so, I would urge them to avoid such terminology. But if not, this needs clarification.

> For all the depths, the accuracy on both the train and test sets peak during the first descent phase.

Given that train and test accuracy plateau seemingly within ~10 gradient steps, I am confused why the models are trained to 100,000 gradient steps. Moreover, most fluctuations and interesting patterns of LC occur beyond ~100 gradient steps (typically past the point that accuracy has saturated), leading me to be skeptical why we are looking at such extremely late training dynamics.

> Figure 3:

Why is the training LNL so spikey but the test LNL so smooth?

> Figure 4: Effect of Batch Normalization on Double Descent

I see no divergence of the test loss near an interpolation threshold. What does this have to do with double descent? And again, double descent is a phenomenon typically described with data to parameters, not gradient steps.

## Minor:

> “ As such, as pose the following question,”

Add a subject to this sentence e.g.,  “As such, we pose the following question,”

> which naturally arise form the

“form” should be “from”

**Questions:**

Most of my equations are integrated into Weaknesses. To extract a few:

1. Is Equation 3 a definition? What is the $\partial \Omega$ notation?

2. What is the parameter $r$?

3. Is SplineCAM an exact method? If so, why does it only work in 2D?

4. Broadly, what do you mean by double descent?

---

### Official Review · Reviewer_n8fX · 2023-11-02

**Soundness:** 3 good
**Presentation:** 3 good
**Contribution:** 2 fair
**Rating:** 5
**Confidence:** 4

**Summary:**

This work builds upon the Max Max/Affine Spline line of works from Balastriero/Baranuik. It shows that the partitions derived from those ideas can be used to calculate summaries that exhibit double descent phenomena during training dynamics. Experimental evaluations are presented on CNNs on MNIST, CIFAR type datasets. Grokking and batch normalization effects are also discussed.

**Strengths:**

1. The overall rationale of the work is clear. Deriving a better characterization of the training dynamics beyond inspection of the loss is generally valuable. The paper shows that repurposing the affine spline formulation does show a double descent behavior.

2. The experiments are satisfactory:  a number of different architectures and datasets are used to demonstrate that the local partition complexity summary is useful.

3. On the practical side, some of the heuristics proposed here make splinecam calculations faster and GPU friendly.

**Weaknesses:**

1. While I appreciate the main double descent observation and some of the experimental findings supporting it, I find the extent of daylight between this work and some of the published results a weakness. The paper does acknowledge clearly that the development will rely on the affine spline. This is not a problem, but except the double descent observation, much of the development described here veers too close to the splinecam paper at CVPR 2023. It is not obvious whether to view this paper as an empirical evaluation of the splinecam work in the context of double descent. Even several figures from that work appear in this submission directly.

2. If the technical adjustments and modifications to the splinecam work are significant, it would be desirable to identify this more explicitly in the text to help the readers assess the distinction clearly.

3. In the absence of new theoretical findings, the paper would also be better served by a deeper analysis of some of the settings mentioned in passing in the Introduction (e.g., self-supervised). What does the proposed characterization of training dynamics reveal in that setting?

**Questions:**

1. Can this work be viewed as applying splinecam with some moderate adjustments, and deriving a summary to check double descent behavior?

---

### Official Review · Reviewer_4pKG · 2023-11-03

**Soundness:** 2 fair
**Presentation:** 2 fair
**Contribution:** 2 fair
**Rating:** 3
**Confidence:** 3

**Summary:**

The authors propose a method to measure the complexity of deep neural network classifiers. In particular, the measure approximates around any given input the number of convex regions that the ReLU activation induces in the input space. Empirical results demonstrate the behavior of the proposed measure in several settings of deep classifiers.

**Strengths:**

- The proposed measure seems to be useful for understanding the local complexity of the classifier while being simple and general enough to be used under different settings.
- Many empirical results about the behavior of the measure in different settings.

**Weaknesses:**

- I think that the writing of the paper can be improved, as in many parts it was quite hard to understand the actual information. Also, in some figures, some extra information is necessary, for example, in Fig. 6 what is the difference in each panel?
- Even if the proposed measure seems to capture the local complexity of the classifier, apart from some connections to related works, I think that a thorough analysis is missing.

**Questions:**

I think that the proposed measure is an interesting and simple approach to approximate the number of convex regions (in some sense the complexity) of the classifier around the training data. However, I do not understand how it should be used and how it helps to analyze the actual behavior of the deep classifier.

---

> ### Author Response · Authors · 2023-11-23
> **Author Response Part 1**
>
> We thank the reviewer for their positive comments, especially on the simplicity of our approximation method and our empirical experiments. Below we respond to individual points raised by the reviewer.
>
> > I think that the writing of the paper can be improved, as in many parts it was quite hard to understand the actual information.
>
> We apologize for the lack of clarity especially Sec. 3.1 and Sec. 4.1 which are integral to understanding how the proposed method works.
>
> To summarize, in this paper we provide a method to approximate the local complexity (LC) of DNNs and show that for a wide range of training settings, the local complexity exhibits an epoch-wise double descent phenomenon. To compute local complexity around any given point $x$ in the input space, we perform the following steps.
> 1. First we start by sampling $P$ orthonormal vectors $B=${$ v_p : p=1...P$}.
> 2.  We use $B$ to denote vertices of a local neighborhood $V$ around any data point $x$ s.t. $V=${$x \pm rv_p : p = 1…P$} where $r$ is a radius parameter. $x$ is therefore the centroid of $V$ consisting of 2P vertices each $r$ distance away from $x$. We define the convex hull $conv(V)$ as the local neighborhood of $x$ in the input space. The dimensionality and volume of the neighborhood can be controlled via $P$ and $r$. Each neighborhood can be considered a non-axis aligned unit $\ell_1$-norm ball scaled by radius $r$.
> 3. Given an input space neighborhood $conv(V)$, we embed $V$ to the input space of layers $\ell$ of the network which precede a non-linearity. We denote the embedded vertices in the input space of layer $\ell$ as $V^\ell$.
> 4. For a given embedded neighborhood $conv(V^\ell)$, we check if there is a change in the pre-activation sign of any given neuron between the vertices. If there is, the neuron (hyperplane) intersects $conv(V^\ell)$. We repeat this for all neurons in all layers and count the total number of intersections as the local complexity.
>
> > Some extra information is necessary, for example, in Fig. 6 what is the difference in each panel
>
> We apologize for the lack of clarity in Fig 6. From left to right each panel presents the local complexity for train, test and random points. For training LC, the LC during the ascent phase changes from low to high to low again while the number of parameters of the network is increase. We do not see such a phenomenon for LC around the test or random points.
>
> > Even if the proposed measure seems to capture the local complexity of the classifier, apart from some connections to related works, I think that a thorough analysis is missing.
>
> In appendix, we have added empirical results analyzing how the parameters of neighborhood $conv(V)$, namely radius $r$ and dimensionality $P$ affects LC computation. In summary, we provide the following observations:
>
> 1. As we increase P, after a certain point, there is marginal increase in LC with increasing P for the same input space point $x$. In Fig.16 left we show that for a 784 dimensional input space and a randomly initialized MLP with width 100 and depth 18, $P=42$ has marginal difference in LC compared to $P=382$.
>
> 2. As we increase radius $r$, the volume of the neighborhood $conv(V)$ increases. While higher neighborhood volume allows computing LC at coarser scales (Fig.16 right), it can also introduce distortion in the embedded neighborhood reducing approximation quality. In this regard, we compute the eccentricity and diameter of the graph formed via $V^\ell$ which is the layer $\ell-1$ embeddings of vertices $V$.  We use $1000$ training points from CIFAR10 to define the neighborhoods for a trained CNN. We see that for larger $r$ both of the deformation metrics exponentially increase, whereas for $r \leq 0.014$ the deformation is lower and more stable. This shows that for lower $r$ our LC approximation for deeper CNN networks is stable since the neighborhood does not get deformed significantly.

---

> > ### Author Response · Authors · 2023-11-23
> > **Author Response Part 2**
> >
> > >  I do not understand how it should be used and how it helps to analyze the actual behavior of the deep classifier
> >
> > While previous work has clearly motivated the need for methods to compute the local complexity of DNNs, exact computation of complexity measures, e.g., the number of linear regions, has combinatorial complexity and is intractable for large deep networks with arbitrary dimensional input spaces. The following are ways in which our results help understand neural networks:
> > 1. Our method can be used to perform fast approximation of the DNN local complexity and can be scaled up to Imagenet experiments as well. In appendix we have presented the LC computed during Resnet18 training on Imagenet upto a top 1 accuracy of 69%. We can also use LC to compare between trained models, compare between sub-groups within the training/test set to quantify bias.
> > 2. Using our method we have presented a novel observation that based on the phase of training, the local complexity around training points might be increasing or decreasing with training epochs. To the best of our knowledge this is the first observation of this kind. This raises new questions regarding our current theoretical understanding of DNN training dynamics, opening new avenues for exploration.
> > 3. Our results indicate that grokking could be a result of 'partition migration' - a phenomenon that we have presented where during the ending stages of training, layerwise neurons (hyperplanes) start orienting themselves with each other and grouping together.
> > 4. In supplementary materials, we present an animation showing the partition migration phenomenon during training, as was shown in Fig.1. Note that while the visualizations are produced via earlier work on partition visualization, our proposed method allows monitoring the partitions at scale during training, which can be used to probe the input space to find where to look. Partition migration can be used to explain the neural collapse phenomenon in DNN training. Because partitions as well as neurons (black lines in Fig.1) start grouping together, loosing linear independence and causing neural collapse.

---

### Meta-Review · Area_Chair_RGQm · 2023-12-14

**Metareview:**

This paper proposes a new complexity measure of deep (ReLU) network that can be measured during the training. Through numerical experiments, they examined the proposed complexity measure to see how it is correlated to important phenomena (grokking, double-descent, memorization) throughout the training.

Unfortunately, this paper contains several issues.
(1) First, the paper's contribution significantly overlaps with existing works, especially with SplineCam. In that sense, the theoretical contributions are not sufficiently novel. On the other hand, the numerical experiments are not strong and solid to locate this paper as an empirical verification of existing work.
(2) Writing can be much improved. The methodology is not well explained, and the experimental results are not exposed in well organized way. It is difficult to see what is the main contribution of this paper and what is the new insight obtained by the analyzes.
(3) Some notations are not defined. For example, the quantity $r$ is not defined although it is a critical quantity in the experiments.

For these reasons, I cannot recommend acceptance for this paper.

**Justification For Why Not Higher Score:**

As I mentioned in metareview, this paper has less novelty and its writing should be much improved. It cannot be accepted unfortunately.

**Justification For Why Not Lower Score:**

N/A

---

### Decision · Program_Chairs · 2024-01-16

Reject